# Acquisition and carriage of genetically diverse multi-drug resistant gram-negative bacilli in hospitalised newborns in The Gambia

Saikou Y. Bah[1,2,6], Mariama A. Kujabi[2,6], Saffiatou Darboe [2], Ngange Kebbeh[2], Bunja F. K. Kebbeh[2], Abdoulie Kanteh[2], Ramatouille Bojang[2], Joy E. Lawn[3], Beate Kampmann[2,4,5], Abdul K. Sesay[2], Thushan I. de Silva [1,2] & Helen Brotherton [2,3✉]

## Abstract

**Background** This detailed genomic study characterised multi-drug resistant-Gram negative bacilli (MDR-GNB) carriage in neonates < 2 kg and paired mothers at a low-resource African hospital.

**Methods** This cross-sectional cohort study was conducted at the neonatal referral unit in The Gambia with weekly neonatal skin and peri-anal sampling and paired maternal recto-vaginal swabs. Prospective bacteriological culture used MacConkey agar with species identification by API20E and API20NE. All GNB isolates underwent whole genome sequencing on Illumina Miseq platform. Multi-Locus Sequence Typing and SNP-distance analysis identified strain type and relatedness.

**Results** 135 swabs from 34 neonates and 21 paired mothers, yielded 137 GNB isolates, of which 112 are high quality de novo assemblies. Neonatal MDR-GNB carriage prevalence is 41% (14/34) at admission with 85% (11/13) new acquisition by 7d. Multiple MDR and ESBL-GNB species are carried at different timepoints, most frequently *K. pneumoniae* and *E. coli*, with heterogeneous strain diversity and no evidence of clonality. 111 distinct antibiotic resistance genes are mostly beta lactamases (*Bla-AMPH, Bla-PBP, CTX-M-15, Bla-TEM-105*). 76% (16/21) and 62% (13/21) of mothers have recto-vaginal carriage of ≥1 MDR-GNB and ESBL-GNB respectively, mostly MDR-*E. coli* (76%, 16/21) and MDR-*K. pneumoniae* (24%, 5/21). Of 21 newborn-mother dyads, only one have genetically identical isolates (*E. coli* ST131 and *K. pneumoniae* ST3476).

**Conclusions** Gambian hospitalised neonates exhibit high MDR and ESBL-GNB carriage prevalence with acquisition between birth and 7d with limited evidence supporting mother to neonate transmission. Genomic studies in similar settings are required to further understand transmission and inform targeted surveillance and infection prevention policies.

## Plain language summary

Bacteria that are resistant to multiple antibiotics are an important cause of infection and death of newborns in low-resource countries, especially small or premature babies born in hospital settings. How these resistant bacteria are acquired on the skin and in the gut of newborns is not known, particularly whether they are commonly transferred from mothers. We studied the bacteria present in small Gambian newborns and their mothers to understand the type of bacteria, amount of antibiotic resistance, number of newborns and mothers affected and similarity of these bacteria between newborns and their mothers. We found that despite many newborns carrying these bacteria, they are different from those present in mothers. This suggests that the bacteria are acquired from the hospital environment. Our study highlights the importance of developing strategies to identify and reduce the presence of such bacteria in hospitals to reduce their acquisition by vulnerable hospitalised newborns.

[1] The Florey Institute of Host-Pathogen Interactions, Department of Infection, Immunity and Cardiovascular Disease, University of Sheffield, Sheffield, UK. [2] MRC Unit, The Gambia at LSHTM, Atlantic Road, Fajara, The Gambia. [3] Department of Infectious Disease Epidemiology, Faculty of Epidemiology & Population Health London School of Hygiene & Tropical Medicine, London, UK. [4] Department of Clinical Research, Faculty of Infectious & Tropical Diseases, London School of Hygiene & Tropical Medicine, London, UK. [5] Institut fur Internationale Gesundheit and Centre for Global Health, Charite Universitatsmedizin, Berlin, Germany. [6]These authors contributed equally: Saikou Y. Bah, Mariama A. Kujabi. ✉email: helen.brotherton@lshtm.ac.uk

Neonatal mortality remains unacceptably high in many African and Asian countries, accounting for 47% of deaths in children under 5 years[1]. Invasive infections are an important contributor to neonatal deaths, with a high burden in Africa and high relative risk of mortality[2,3]. Small vulnerable neonates born either premature (< 37 weeks gestation) and/or low birth weight (LBW; < 2.5 kg) are at greatest risk of infections due to impaired innate and adaptive immunity[4], prolonged hospital stay, and invasive procedures[5]. Intestinal carriage of pathogens with translocation across the gut wall is associated with late onset infections and inflammatory disorders[6] and premature infants skin integrity is typically impaired, providing an additional route for invasive infection.

An estimated 31% of the 690,000 annual neonatal deaths associated with sepsis are potentially attributable to antimicrobial resistance (AMR)[7]. Within Gram-negative bacteria (GNB), Enterobacterales are the leading cause of severe bacterial infections in African neonates[2,8,9], with *Klebsiella pneumoniae* and *Escherichia coli* most commonly implicated. Multi-drug resistance (MDR) is seen in up to 82% of invasive neonatal GNB in Africa[10,11], with prevalence increasing[11] and management challenges due to limited diagnostics and therapeutic options[7]. Extended Spectrum Beta Lactamase (ESBL)-producing GNB were listed as pathogens of high priority for research and antibiotic development by the World Health Organisation in 2017[12], and represent a neonatal public health emergency which is critical to address if global targets to reduce neonatal mortality to ≤12/1000 livebirths are to be met by 2030[13].

Neonatal MDR-GNB carriage is associated with invasive blood stream infections[14–16], yet detailed understanding of how neonates acquire MDR-GNB within hospitals in resource limited settings (RLS) is limited. LBW[17], prolonged hospital stay and antibiotic use are risk factors[18] for neonatal MDR-GNB acquisition and an association between neonatal ESBL-GNB carriage and premature delivery has been reported[19]. In addition, the intestinal colonisation pattern of hospitalised premature neonates differs from that of healthy, term, breastfed infants but there is a paucity of gestational age specific data from the lowest resource settings and most data originates from HIC settings where infection prevention control and health system context differs. Environmental sources of MDR-GNB on African neonatal units have also been described[20], with contaminated fluids, antibiotic vials, equipment and surfaces implicated and linked to outbreaks[21,22]. Maternal colonisation is a well-recognised risk factor for neonatal acquisition and infection with Gram-Positive bacteria such as *Group B Streptococcus*[23]. However, the role of maternal colonisation in neonatal MDR-GNB acquisition, especially in Africa, has not undergone rigorous scrutiny[24], and the relative contribution of vertical versus horizontal transmission is not known[25]. This is an important gap to address for development of targeted infection prevention control strategies to reduce neonatal MDR-GNB carriage and subsequent invasive infections.

This study aimed to characterise MDR-GNB carriage in small vulnerable newborns at a low-resource African neonatal unit (NNU), with exploration of acquisition in relation to maternal carriage using whole genome sequencing (WGS). Objectives included: 1) To determine species specific MDR-GNB and ESBL-GNB carriage prevalence for neonates and paired mothers; 2) To describe strain-specific *K. pneumoniae* and *E. coli* carriage; 3) To describe antibiotic resistance genes and 4) To explore relatedness of *K. pneumoniae* and *E. coli* isolates within newborn-mother dyads.

In summary, we identified high carriage prevalence of MDR- and ESBL-producing GNB for small vulnerable neonates within 24 h of NNU admission and extensive acquisition after 7d of hospital stay. Multiple MDR and ESBL-GNB species are present in individual neonates at different time points, most commonly *K. pneumoniae* and *E. coli*. There is heterogeneous strain diversity, no evidence of clonality and a wide range of AMR genes, most commonly beta-lactamases. Maternal carriage prevalence of MDR-GNB, predominantly *E. coli*, is very high. However, only one newborn-mother dyad have evidence of genetically identical strains for both *K. pneumoniae* and *E. coli*. These results suggest that multiple environmental sources play an important role in neonatal acquisition in this setting, but further genomic studies are required to fully understand transmission and inform targeted surveillance and infection prevention policies.

## Methods

**Study design**. This cross-sectional cohort study was conducted from April to August 2017 as part of a feasibility study to inform the design of a clinical trial investigating the effect of early kangaroo mother care (KMC) on survival of small vulnerable newborns[26].

**Study setting**. Recruitment took place at the national neonatal referral unit in The Gambia, (Edward Francis Small Teaching Hospital (EFSTH), Banjul). Approximately 1400 neonates are admitted per year[27] from a mixed in-born ( ~ 6000 births/year in hospital) and out-born (other health facilities or home) population. Neonatal mortality in The Gambia declined from 49 to 26 per 1000 live births between 1990 and 2018 [1], but is still substantially higher than the SDG 3.2 target of 12 per 1000 live births. 12% of Gambian neonates are born preterm[28], 17% LBW[1] and 28% of neonatal deaths are due to infections[29], with likely underestimation of the contribution of infection to mortality of small newborns.

The inpatient case fatality rate at EFSTH ranged from 35% (all admissions) to 48% for neonates <2 kg from 2010 to 2014, with prematurity or LBW accounting for 27% of all admissions[27]. At the time of this study, WHO level 2 newborn care was provided with oxygen via concentrators, phototherapy, access to blood transfusion and intravenous (IV) fluids. Empirical first line antibiotics were IV ampicillin and gentamicin (age < 72 h), with ceftriaxone and flucloxacillin used for both community onset and suspected hospital acquired infections (HAI) (age ≥ 72 h). Ciprofloxacin was second line treatment with carbapenems rarely available. Post-mortems, blood cultures, C-reactive protein and other infection bio-markers were not available, hence the prevalence and contribution of sepsis to mortality in this cohort is not known. Admission rates ranged from 80 to 100 neonates/month during the quieter dry season (January–August) to 140–160 neonates/month during the rainy season (September–December)[27].

**Study population**. All neonates ≤ 2 kg admitted during the study period (April–July 2017) were screened for eligibility with inclusion criteria: weight < 2 kg and admission age < 24 h. Neonates were excluded if death occurred before or during screening or in the absence of informed consent. Specific exclusion criteria for skin swabs included topical antibiotics or steroids applied to skin since birth, and generalised or local skin disorder within 4 cm of swabbing site[30]. Exclusion criteria for peri-anal swabs included imperforate anus or anal stenosis, previous gastrointestinal surgery or diarrhoea within preceding 24 h[30]. Mothers were approached for consent to provide recto-vaginal (RV) swabs once they were available, with exclusion if known HIV infection, major gastrointestinal surgery within previous 5 years, diarrhoea or constipation within preceding 24 h, or current sexually transmitted infection[30]. Participant identification was pseudo-

anonymised using unique identification numbers, with identity known only by the researchers.

**Data collection & procedures**. Neonatal swabs, anthropometric and clinical data, gestational age assessment (New Ballard score[31]), antibiotic use and in-hospital outcomes were collected via direct observation, parental interviews and medical record review as soon as possible after admission then weekly (day 7, 14, 21, 28) until discharge or death. Data were recorded electronically using REDCap™. Neonatal skin swabs were obtained by composite sampling from the xiphisternum and peri-umbilical area. Peri-anal samples were taken in lieu of stool or rectal samples for intestinal carriage[32] due to less invasive sampling. RV samples were taken from consented mothers as a combined swab. All samples were taken by trained personnel with FLOQ® swabs and stored in Amies transport medium at +4 to +8 °C prior to transfer to the Medical Research Council, The Gambia at London School of Hygiene & Tropical Medicine (MRCG), Fajara, The Gambia, within 48 hrs.

**Microbiological processing**. All prospective bacteriological processing and storage of isolates occurred at MRCG, ISO-15189 accredited Clinical Laboratories. Samples were processed immediately by culture on MacConkey agar plates incubated aerobically at 35–37 °C, with identification of Enterobacterales and other GNB using API 20E and API 20NE respectively.

**Molecular processing and bioinformatic analysis**. Whole genome sequencing was performed at the MRCG Genomics Facility in The Gambia. Deoxyribonucleic acid (DNA) was extracted from cultures of all identified GNB, expanded from single bacterial colonies using the QIAamp DNA extraction kit (Qiagen) following manufacturer's instructions and Illumina Miseq sequenced to 250 cycles. Quality control and trimming of raw sequence reads were done using FastQC (v0.11.8) and trimmomatic (v0.38) respectively to remove low quality bases and sequencing adapters[33]. Whole genome de novo assemblies were generated using SPAdes (kmers: 21,33,55 and 77) and quality checked using Quast[34,35]. All draft assemblies greater than 500 contigs were removed from downstream analyses. Multi locus sequence typing (MLST) was done using the get_sequence_type from the mlst_check from the Sanger institute pathogen group https://github.com/sanger-pathogens/mlst_check). Genomes were annotated using Prokka and core genomes analysed using Roary[36,37]. Single nucleotide polymorphisms (SNP)-distance was used to calculate genetic distances between isolates of the same species to infer relatedness. ABRicate was used to determine AMR gene carriage using the ARG-ANNOT database setting a minimum coverage of 70% and identity of 75% (https://github.com/tseemann/abricat). Genotypic MDR was defined as presence of AMR genes encoding for three or more different antimicrobial classes, as per MEGARes database[38]. Isolates were defined as ESBL producing as per updated Bush-Jacoby functional classification system[39–41] if ≥ 1 previously described ESBL gene type was identified, regardless of whether MDR was present. Maximum likelihood phylogenetic trees were generated from aligned core SNPs using RAxML with 100 bootstraps, visualized and annotated in iTOL[42,43].

**Statistics and reproducibility**. Details regarding reproducibility of the molecular processing and bioinformatic analyses are outlined in the relevant sections of the methods. Carriage prevalence rate was calculated as the proportion of participants colonised with a genotypically identified bacteria out of total number of sampled participants, with stratification by MDR and ESBL

status, type of participant (neonate/mother), site of neonatal sampling (skin/peri-anal), and day of sample collection. Participant characteristics were described according to data distribution with complete case analysis for missing data. As this was an observational pilot study no sample size was calculated a-priori.

**Ethical approvals**. All relevant ethical regulations were followed, with ethical approval granted from LSHTM Observational Ethics Committee (Ref. 11887) and the Gambian Government/MRCG Joint Ethics Committee (Ref. 1503). Written informed consent was sought from the neonates' first available relative prior to data collection with separate consent for maternal sampling sought from the mother. Consent was sought for future research on samples with exclusion from genomic analysis if not provided. Participants were free to withdraw from the study at any time[26].

**Inclusion statement**. This research included local researchers in study design, implementation, interpretation and authorship. The research is relevant in The Gambia and as determined by local partners. Roles and responsibilities for sample processing and analysis were agreed ahead of genomic activities with capacity building for local researchers (MK) incorporated. All samples were processed in The Gambia with bioinformatic analyses conducted by Gambian team members. Relevant local and regional research findings were considered in this manuscript and cited appropriately.

**Reporting summary**. Further information on research design is available in the Nature Portfolio Reporting Summary linked to this article.

## Results

**Enrolment & study participants**. Of 89 neonates screened, 36 met eligibility criteria and 34 underwent sampling with 114 neonatal carriage swabs obtained (Fig. 1). Twenty-one mothers were sampled, of which 19 were linked to sampled neonates. 21 neonate-mother dyads were included due to presence of two twin-mother pairs. 76% of maternal RV samples were taken within 24 h of neonatal admission (Table 1).

The median neonatal admission weight was 1330 g (71% < 1.5 kg) and median gestational age 33 weeks, with 18% (6/34) twin pairs. 91% (32/34) were born in a health facility with a combination of neonates born at the study site (inborn) and elsewhere (out-born) with postnatal transfer. At least one sepsis risk factor (maternal fever, suspected chorioamnionitis or prolonged rupture of membranes >18 h) was present for 13% (3/23) of neonates with 17% (4/23) receiving antibiotics within 48 h before delivery. The inpatient case fatality rate within 28 postnatal days was 62% (21/34), with median age at death 2.5 days (Table 1). Reliable data on the cause of death was not available.

**Sampling and overview of Gram-Negative Bacilli isolates**. 135 neonatal and maternal carriage swabs yielded 137 GNB isolates from conventional bacteriology with 112 high quality de novo assemblies obtained (Fig. 1).

**Spectrum of Gram Negative Bacilli carriage and rates of genotypic multi-drug resistance**. Of 112 high quality de novo assemblies obtained, 70% (78/112) originated from neonates and 30% (34/112) from mothers (Supplementary Fig. 1a). *E. coli* (40%, 45/112) and *K. pneumoniae* (33%, 37/112) were the most frequently identified species. Half (23/45) of the *E. coli* isolates were derived from maternal RV samples. 76% (28/37) of the *K. pneumoniae* identified were neonatal in origin, predominantly

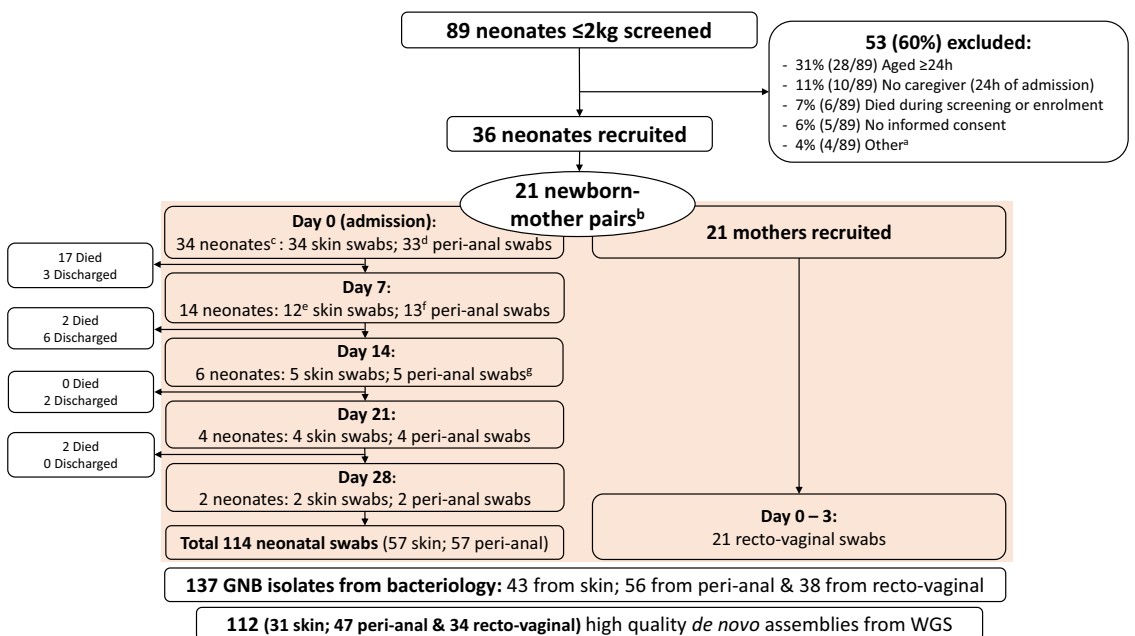

**Fig. 1 Overview of screening, enrolment, carriage sampling and isolate sequencing for neonates and mothers.** [a]Other reasons for non-recruitment included weight >2 kg on study scales (*n* = 2) and no staff available to perform study procedures (*n* = 2). [b]Two mothers were sampled in absence of neonatal paired swabs and two sets of twins were included. [c]36 neonates were enrolled but 2 were not sampled, due to rapid deterioration and death (*n* = 1) and lack of consent for neonatal sampling (*n* = 1). [d]One participant did not meet eligibility for peri-anal samples due to imperforate anus. [e]Two neonates did not have skin swabs taken: One met exclusion criteria; One had consent withdrawn for repeat sampling. [f]One neonate did not have peri-anal swabs taken as consent was withdrawn for repeat sampling. [g]One newborn did not have skin or peri-anal samples taken due to error. GNB Gram-Negative Bacilli, WGS Whole Genome Sequencing.

from peri-anal samples (Supplementary Fig. 1a). Nearly three-quarters of all GNB isolates (73%, 82/112) exhibited genotypic MDR including 76% (28/37) of *K. pneumoniae* and 73% (33/45) of *E. coli* (Supplementary Fig. 1b).

**Neonatal & maternal MDR-Gram Negative Bacilli carriage during NNU admission.** 41% (14/34) of neonates carried ≥1 MDR-GNB and 32% (11/34) ≥1 ESBL-producing GNB at time of NNU admission. MDR-GNB carriage was split equally between skin (9/34, 26%) and peri-anal (8/33, 24%) with three newborns colonised with an MDR-GNB at both sites (Table 2). All surviving neonates carried ≥1 MDR-GNB after one week, with 85% (11/13) colonised with at least one new MDR-GNB compared to admission (Table 2).

29% (10/34) of neonates were colonised with *E. coli* at admission, predominantly on the skin (21%, 7/34), with both MDR and ESBL *E. coli* carriage present in 18% (6/34) of neonates. Comparatively less neonates carried *K. pneumoniae* at admission (21%, 7/34), with only three (9%) colonised with a MDR or ESBL *K. pneumoniae*. The carriage prevalence increased for both MDR and ESBL *E. coli* and *K. pneumoniae* during the first week of admission, with the greatest increase observed for *K. pneumoniae* (9% to 54% for both MDR and ESBL) and to a lesser extent *E. coli* (18% to 23% for both MDR and ESBL) (Table 2). Eleven neonates had 21 isolates identified from samples taken after day 7 of admission, mostly *K. pneumoniae* (57%, 12/21 isolates), identified predominantly from peri-anal swabs (Supplementary Fig. 1a).

76% (16/21) of mothers carried ≥1 MDR-GNB recto-vaginally and 62% (13/21) had an ESBL-producing pathogen. *E. coli* was most commonly identified, with 76% (16/21) of mothers colonised with an MDR-*E. coli*. One quarter (24%, 5/21) of mothers had RV carriage of MDR-*K. pneumoniae*.

**Genetic diversity of Gram-Negative Bacilli carriage isolates**
*Klebsiella pneumoniae.* We obtained 37 quality de novo *K. pneumoniae* assemblies. Eighteen different sequence types (STs) were determined. ST607 was the predominant ST (11%, 4/37), followed by ST37, ST133 and ST307 (8%, 3/37 each). 19% (7/37) of isolates were not assigned a ST. Seven neonates had multiple *K. pneumoniae* isolated at different time points, of whom four had distinct STs: N019 (3 isolates), N020 (4 isolates), N029 (3 isolates) and N040 (2 isolates). Only two neonates carried a genetically identical *K. pneumoniae* at two or more different time points: 1) A female neonate carried *K. pneumoniae* ST502 at 20d and 28d (N002; SNP difference=11); 2) A male neonate carried *K. pneumoniae* ST607 at 7d and 14d (N012; SNP distance=0). One set of twins carried the same *K. pneumoniae* (ST37, SNP distance=0) on day 8 (N019, twin 1, peri-anal) and day 21 (N020, twin 2, skin). This twin pair also had peri-anal carriage of an identical *K. pneumoniae* (ST476;SNP distance=17) on day 14 (N019) and day 29 (N020), which was not identified in their mother (M009). Furthermore, there was only a single instance of a newborn-mother dyad with identical *K pneumoniae* (ST3476; SNP distance=0), isolated from a neonatal d0 peri-anal swab (N048) and maternal RV swab (M022)(Fig. 2a). This involved a female preterm singleton, admitted following vaginal delivery at another health centre. This newborn died within 7 postnatal days, hence further samples weren't available.

*Escherichia coli.* 45 high quality *E. coli* genomes were obtained with 21 different STs identified and two isolates (4.4%) not assigned a ST. ST10 was most common (20%, 9/45), followed by ST69, ST127 and ST3580 (8.8%, 4/45 each). The most common neonatal derived *E. coli* strains were ST10 and ST3580 (18%, 4/22, each). Three neonates carried multiple *E. coli* isolates at varying time-points: 1) A female singleton weighing 1500 g had *E coli* on both skin and peri-anal swabs at d0 (ST58 and unassigned ST;

| Table 1 Clinical and socio-demographic characteristics of neonatal participants with carriage samples at time of admission | |
|---|---|
| Clinical or socio-demographic characteristic | N = 34[a] |
| Admission weight (g), median (IQR) | 1330 (1160–1510) |
| Distribution of admission weight, Nº (%) | |
| <1000 g | 6 (18) |
| 1000–1499 g | 18 (53) |
| 1500–1999g | 10 (29) |
| Male sex, Nº (%) | 18 (53) |
| Gestational age (weeks), median (IQR) (n = 30) | 33 (31–34) |
| Distribution of gestational age, Nº (%) | |
| <28 weeks | 0 (0) |
| 28–31 + 6 weeks | 11 (32) |
| 32–36 + 6 weeks | 18 (53) |
| ≥ 37 weeks | 5 (15) |
| Age at admission (h), median (IQR) | 4.0 (1.3–7.3) |
| Maternal sampling | N = 19[b] |
| < 24 h after neonatal admission | 16 |
| 24–48 h after neonatal admission | 2 |
| 48–72 h after neonatal admission | 1 |
| Twin pregnancy, Nº (%) | 6 (18) |
| Place of birth, Nº (%) (n = 23) | |
| Home | 2 (9) |
| Primary health centre | 8 (35) |
| Secondary hospital | 3 (13) |
| Tertiary hospital (EFSTH) | 10 (43) |
| Hygienic cord care at birth[c] No (%) (n = 22) | 20 (91) |
| Maternal antibiotics within 48 h of delivery, Nº (%) (n = 23) | 4 (17) |
| Sepsis risk factors present[d] Nº (%) (n = 23) | 3 (13) |
| Axillary temperature (ºC), median (IQR) (n = 20) | 36.2 (34.7–36.8) |
| Blood glucose (mmol/L), median (IQR) (n = 21) | 5.1 (3.5–6.3) |
| Neonatal IV antibiotics within week 1, Nº (%) (n = 21) | 18 (86) |
| Ampicillin doses in week 1, median (IQR) (n = 18) | 19 (12–23) |
| Gentamicin doses in week 1, median (IQR) (n = 18) | 5 (2–7) |
| Ceftriaxone doses in week 1, median (IQR) (n = 2) | 0.5 (0–1) |
| Maintenance IV fluids within week 1, Nº (%) (n = 21) | 19 (91) |
| Maternal expressed breast milk within first week, Nº (%) (n = 21) | 16 (76) |
| Inpatient outcome (died), Nº (%) | 21 (62) |
| Died between 0 and 7d, Nº (%) | 17 (50) |
| Died between 7 and 14d, Nº (%) | 2 (6) |
| Died between 14 and 21d, Nº (%) | 0 |
| Died between 21 and 28d, Nº (%) | 2 (6) |
| Age at death (h), median (IQR) (n = 21) | 60 (9–119) |

[a]Denominator 34 unless otherwise stated.
[b]21 mothers gave consent and were sampled but only 19 were linked to neonates with paired samples. This was due to demise of a newborn (n = 1) and lack of consent (n = 1). This comprised 21 mother-newborn dyads due to two twin pairs in the sampled cohort.
[c]Hygienic cord care is defined as the umbilical cord tied with sterile clamp and/or nothing was applied to the cord.
[d]Sepsis risk factors defined as premature rupture of membranes >18 h, maternal fever within 48 h of delivery, offensive smelling liquor or chorioamnionitis.
d days, EFSTH Edward Francis Small Teaching Hospital, h Hours, IQR Interquartile range.

N014); 2) A female twin carried E. coli ST10 on day 0 (PA) and day 7 (skin and PA)(SNP distances=1–128; N019) and E. coli ST127 was present on day 14 (PA); 3) A male singleton had four identical isolates of E. coli ST3580 (SNP distance=0-3; N029) on skin and peri-anal samples taken between day 7 and day 21, with E. coli ST648 also present on day 21 (Fig. 2b). E. coli ST10 was the most commonly observed isolate in maternal RV samples (24%,

5/21). 14% (3/21) of mothers carried >1 E. coli ST. Only one newborn-mother pair had identical E. coli carriage (ST131; SNP distance=0), with neonatal peri-anal carriage at d0 (N048) and maternal RV sample (N022) which was obtained within 24 h of admission. This mother-neonate pair also had identical K. pneumoniae strains, as described above.

**Antimicrobial resistance (AMR) gene carriage**. A total of 1131 AMR genes were identified from 112 isolates, representing 111 distinct gene types, and encoding resistance for 10 antibiotic classes. Beta-lactam resistance was the most common (43%, 48/111), followed by resistance to Aminoglycosides (18%, 20/111) (Fig. 3, Supplementary Data 1). All K. pneumoniae and E. coli isolates had ≥ 2 AMR genes, with a median of 11 AMR genes per isolate for both bacteria (K. pneumoniae range (3-19)); E. coli range (5-15). An ESBL-gene was present in 59% (66/112) of all GNB isolates (Supplementary Fig. 1c).

K. pneumoniae isolates had a total of 446 AMR genes (51 distinct gene types), encoding resistance to 10 antibiotic classes, most commonly beta-lactamases (35.9%, 160/446). Nineteen distinct beta-lactamase genes were identified, most commonly BlaAmpH (97%, 36/37 isolates) and BlaPBP (94%, 35/37 isolates). At least one ESBL gene was present in 70% (26/37) of K. pneumoniae isolates and 89% (25/28) of MDR-K. pneumoniae, most frequently CTX-M-15 (20/37, 54%) and Bla-TEM-105 (18/37, 49%) (Fig. 3, Supplementary Data 1). Aminoglycoside resistance was present in 73% (27/37) of isolates with AGlyStrB (21/37, 57%), AGlyStrA (20/37, 54%) and AGlyAac3-IIa (18/37, 49%) most common (Supplementary Data 1).

E. coli isolates harboured 456 AMR genes (32 distinct gene types), encoding resistance to 8 different antibiotic classes, most commonly beta-lactamases (44.1%, 201/456). Eight types of beta-lactamase genes were present, with BlaAmpH, BlaPBP and BlaAmpC2 carried by all isolates along with BlaAmpC1 in 78% (35/45). Over two-thirds (69%, 31/45) of E. coli isolates harboured an ESBL-gene (Supplementary Fig. 1c), mostly Bla-TEM-105 (53%, 24/45) (Fig. 3, Supplementary Data 1). Aminoglycoside resistance was present in 73% (33/45) of E. coli isolates, predominantly due to AGlyStA (31/45, 69%) (Supplementary Data 1).

None of the major carbapenemase resistance genes (VIM-, IMP- and NDM-type metallo-beta-lactamases, KPC- nor OXA-48) were identified in any GNB. However, 91% (10/11) of A. baumannii isolates harboured blaMbl, a class B3 beta-lactamase with carbapenemase activity (Fig. 3, Supplementary Data 1).

## Discussion

Our observed high rate of inpatient neonatal MDR-GNB acquisition (85%) is greater than in European outbreak situations (24% in Norway[44]), middle-income NNUs such as Morocco (58%)[21] and Malaysia (22% and 52%)[45]. Our findings are comparable to phenotypic data from Ethiopia (74% prevalence of ESBL-GNB after 48 h of admission)[46] and other West African countries such as Ghana (65%)[47], although the proportion of ESBL-GNB varies considerably within African regions and hospitals, as shown by large heterogeneity in pooled prevalence from East Africa (12–89%)[48]. Our cohort of mixed inborn and out-born neonates had high MDR-GNB carriage prevalence at time of NNU admission (41%), suggesting that rapid colonisation occurs during the pre-admission period, although we are unable to comment on precise timing and source of acquisition. Our observed 54% MDR K. pneumoniae carriage prevalence after 7 days of admission is similar to that reported from a tertiary NNU in Ghana in which 49.6% of neonates had phenotypic MDR K. pneumoniae and 75.6% exhibited ESBL activity at median 3d of

**Table 2 Genotypic MDR-GNB and ESBL-GNB carriage amongst neonates and mothers from admission to day 7 of hospitalisation.**

| | Neonates | | | | | | Mothers |
|---|---|---|---|---|---|---|---|
| | Total[a] | | Peri-anal | | Skin | | Recto-vaginal |
| | D0 N = 34 | D7 N = 13 | D0 n = 33 | D7 n = 13 | D0 n = 34 | D7 n = 12[b] | D0 – D3 n = 21 |
| ≥1 GNB isolated, Nº (%) | 19[c] (56) | 13[d] (100) | 12 (36) | 12 (92) | 15 (44) | 5 (42) | 17 (81) |
| ≥1 MDR-GNB, Nº (%) | 14[c] (41) | 13[d,e] (100) | 8 (24) | 12 (92) | 9 (26) | 4 (33) | 16 (76) |
| ≥1 ESBL-GNB, Nº (%) | 11[c] (32) | 9[d,f] (69) | 5 (15) | 9 (69) | 7 (21) | 3 (25) | 13 (62) |
| *Escherichia coli*, Nº (%) | 10[c] (29) | 3[d] (23) | 4 (12) | 3 (23) | 7 (21) | 1 (8) | 17 (81) |
| MDR-*E. coli*, Nº (%) | 6 (18) | 3[d] (23) | 3 (9) | 3 (23) | 3 (9) | 1 (8) | 16 (76) |
| ESBL-*E. coli*, Nº (%) | 6 (18) | 3[d] (23) | 3 (9) | 3 (23) | 3 (9) | 1 (8) | 12 (57) |
| *Klebsiella pneumoniae*, Nº (%) | 7[c] (21) | 8[d] (62) | 4 (12) | 7 (54) | 3 (9) | 2 (17) | 8 (38) |
| MDR-*K. pneumoniae*, Nº (%) | 3[c] (9) | 7[d] (54) | 1 (3) | 6 (46) | 2 (6) | 2 (17) | 5 (24) |
| ESBL-*K. pneumoniae*, Nº (%) | 3[c] (9) | 7[d,f] (54) | 1 (3) | 6 (46) | 2 (6) | 2 (17) | 3 (14) |
| *Acinetobacter* spp., Nº (%) | 5[g] (15) | 2[g] (15) | 4 (12) | 2 (15) | 4 (12) | 1 (8) | 0 |
| MDR-*Acinetobacter* spp. Nº (%) | 1 (3) | 2 (15) | 1 (3) | 2 (15) | 0 | 0 | 0 |
| ESBL-*Acinetobacter* spp. Nº (%) | 0 | 0 | 0 | 0 | 0 | 0 | 0 |
| Other GNB[h] Nº (%) | 8 (24) | 3 (23) | 3 (9) | 3 (23) | 6 (18) | 0 | 2 (10) |
| MDR-Other, Nº (%) | 7 (21) | 3 (23) | 3 (9) | 3 (23) | 5 (15) | 0 | 2 (10) |
| ESBL-Other, Nº (%) | 4 (12) | 2 (15) | 1 (3) | 2 (15) | 3 (9) | 0 | 0 |

[a]Neonates with either peri-anal or skin sample.
[b]1 neonate underwent peri-anal sampling at day 7 but met clinical exclusion criteria for skin sampling.
[c]8 neonates had a GNB identified from both peri-anal and skin samples at first sampling, 3 with a MDR-GNB at both sites and 1 with ESBL-GNB at both sampling sites.
[d]4 neonates had a GNB identified from both peri-anal and skin samples at day 7, with 3 neonates carrying MDR and ESBL pathogens at both sampling sites.
[e]85% (11/13) of surviving neonates who were sampled acquired a new MDR-GNB by day 7 of admission with 12 isolates identified (*E. coli* n = 2; *K. pneumoniae* n = 6; *A. baumannii* n = 2; *E. cloaceae* n = 1; *Pseudomonas aeruginosa* n = 1).
[f]69% (9/13) of surviving neonates who were sampled acquired a new ESBL-GNB by d7 of admission.
[g]3 neonates had *A. baumannii* present on skin and peri-anal swabs.
[h]Other GNB: Enterobacter cloacae (n = 5); Citrobacter freundii (n = 2); Cronobacter sakazakii (n = 3); Pseudomonas aeruginosa (n = 2); Pseudomonas putida (n = 1); Salmonella enterica (n = 1).
*ESBL* Extended Spectrum Beta Lactamase, *GNB* Gram-Negative Bacilli, *MDR* Multi-drug resistant.

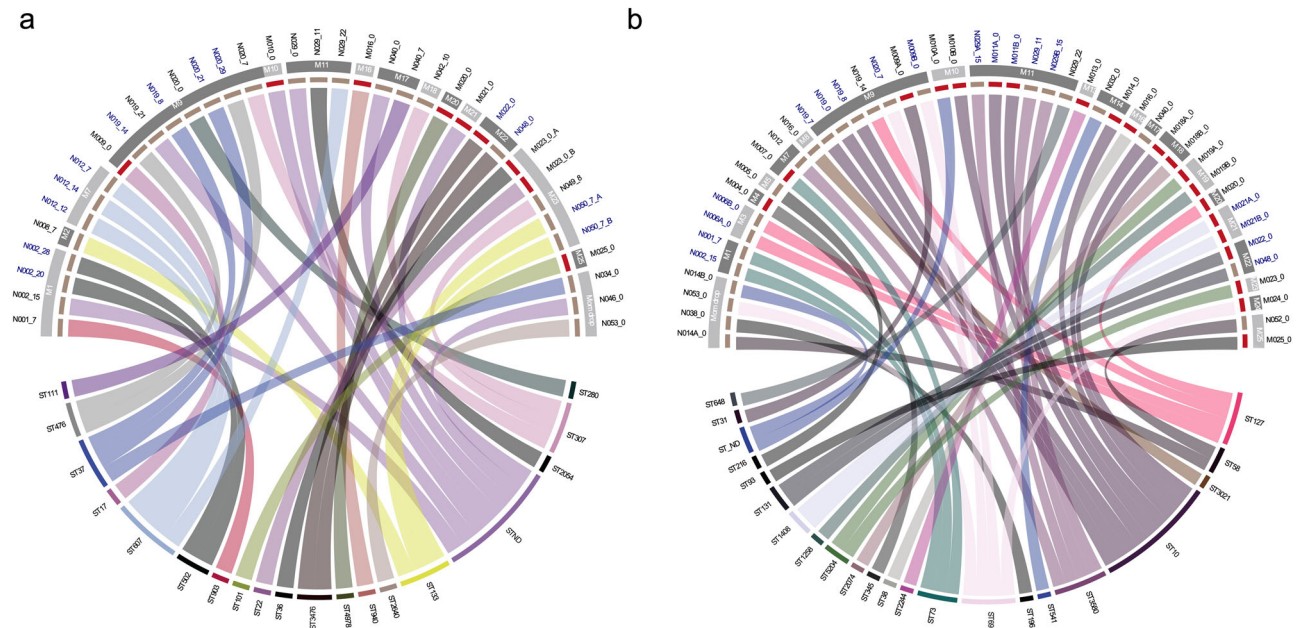

**Fig. 2 Multi-locus sequencing types and genetic relatedness of *K. pneumoniae* and *E. coli* identified from paired newborn-mother carriage samples.** **a** *Klebsiella pneumoniae*. **b** *Escherichia coli*. SNP-distance was used to determine genetic relatedness of isolates using the core-genome alignment obtained from Roary. Sequence type (ST) were determine using the mlst package https://github.com/tseemann/mlst. SNP distances were imported into R to generate the circus plots using the circlize package. In the top half of the circus plot, the inner segments indicate whether the isolates were collected from a neonate (brown) or mother (red), mothers' study ID (MXX) and day of sampling labelled on the outside separated by underscore (_). Sample IDs highlighted with the blue font are neonate-mother pairs that have the ST. The bottom half indicates the ST types of *K. pneumoniae* or *E. coli*. Connecting lines joining the upper and lower halves of the circus plot indicates to which ST a particular isolate belongs.

admission[18]. The high carriage rates at both admission and after 7 days likely reflects varying health system factors such as availability of WASH resources and optimal hand hygiene, provision of sterilisation techniques, over-crowding and under-staffing[49], sub-optimal infection prevention control systems and re-use of disposable consumables and equipment. All these factors have been linked to MDR-GNB outbreaks in other African NNUs[20] and were also observed previously at this site[22].

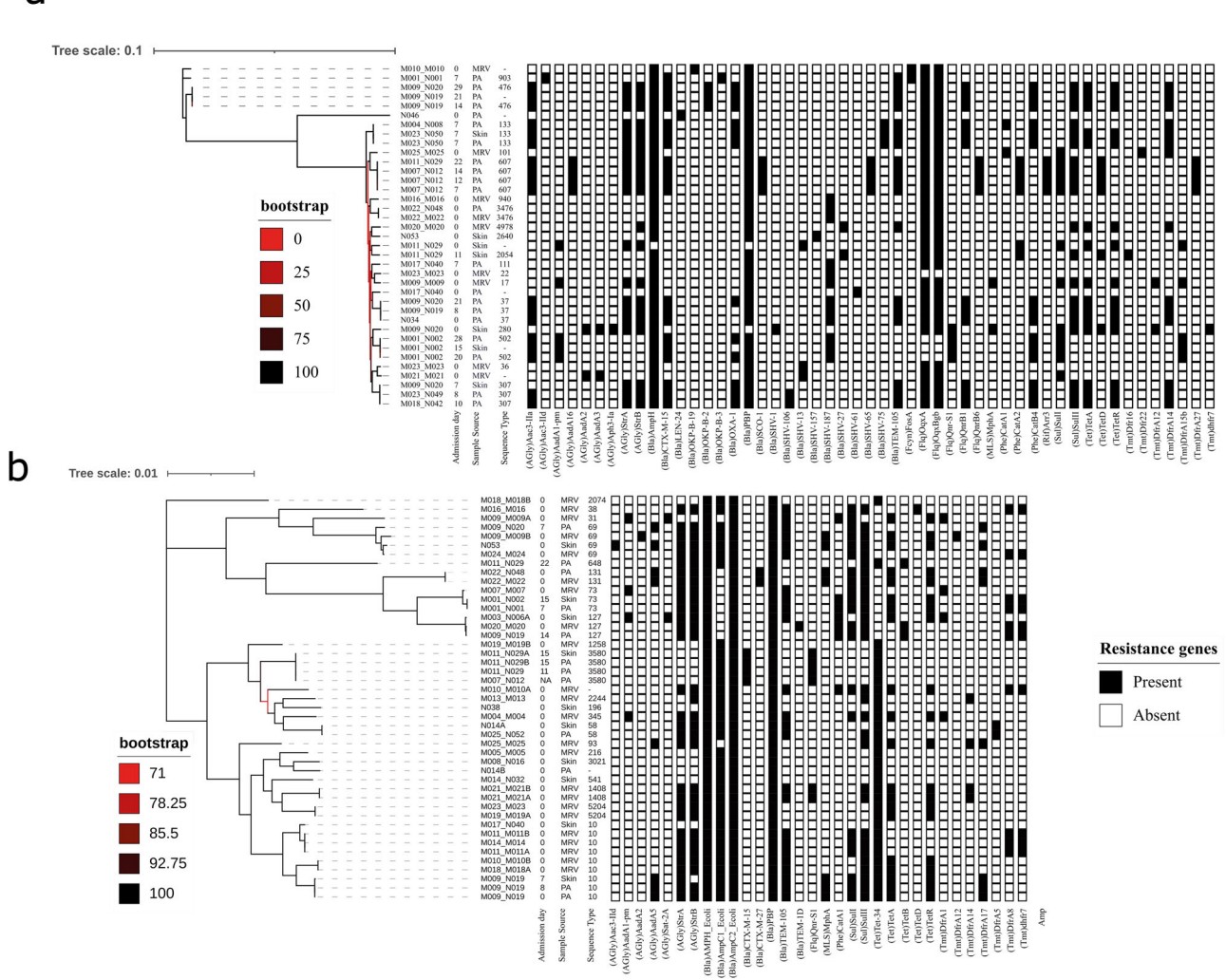

**Fig. 3 Phylogenetic trees of *K. pneumoniae* and *E. coli* isolates identified from paired newborn-mother carriage samples. a** *Klebsiella pneumoniae*.
**b** *Escherichia coli*. A maximum likelihood phylogenetic tree was constructed from core-genome SNPs using RAxML with 100 bootstraps. For both species isolates are clustered by STs. Also shown are the presence (black-filled square) or absence (white empty-square) of antimicrobial resistance genes (AMR).

MLST identified substantial intraspecies diversity with 21 different *E. coli* sequence types (most frequently ST10, ST69, ST127 and ST3580), and 18 *K. pneumoniae* strains (most commonly ST607, ST37, ST133, ST307). *E. coli* strains range from harmless commensals to pathogenic variants associated with invasive infections[50] and several strains colonising our cohort have been linked to neonatal infections in Africa and Asia (ST10, ST69)[50]. *E. coli* ST131 is a highly virulent strain and major neonatal pathogenic variant[50] yet was carried by only one neonate and paired mother in our cohort. This low carriage prevalence is consistent with a community study in Guinea Bissau which identified ST131 in 4% of *E. coli* isolates from >400 children, including neonates[51]. The ST131 *E. coli* strains within the dyad were identical, suggesting mother to newborn transmission had occurred, which may have relevance for settings in which *E. coli* 131 is a more dominant carriage strain.

Two of our most frequently identified *K. pneumoniae* strains (ST37, ST307) are associated with invasive neonatal infection and have been previously reported from Ethiopia (ST37), Rwanda (ST307) and Nigeria (ST307)[50]. We did not identify any *K. pneumoniae* ST39 or ST31535 (*K. quasipneumoniae*), which were both implicated in contemporaneous outbreaks at the site one year previous to our sampling period[22], suggesting that prior outbreaks were contained. There are limited other African genomic data describing strain specific neonatal *E. coli* and *K. pneumoniae* carriage and this is a priority research area.

Beta-lactamases were the most common AMR genes identified, with a predominance of *AmpH*, *penicillin binding protein (PBP)*, *TEM* and *CTX-M* gene types, conferring resistance to ampicillin and 3rd generation cephalosporins, which, along with Gentamicin, are WHO recommended first and second line antibiotics for neonatal sepsis. The low prevalence of carbapenem resistance genes in our isolates contrasts with higher levels reported from Kenya (14%)[52], Ghana (15.6%)[53], and Thailand (64%)[54], likely reflecting the limited availability of carbapenem antibiotics in our setting and thus reduced selective pressure. However, nearly ubiquitous presence of *blaMbl* in *A. baumannii* is of concern due to risk of inter-species transfer due to mobile genetic elements[55]. This is an area of high priority for future genomic surveillance in The Gambia and elsewhere in West Africa to help guide antimicrobial stewardship and AMR surveillance.

The high maternal MDR-GNB carriage prevalence (76%) is consistent with some other African studies[24], although is markedly higher than a similar neonatal-maternal dyad study in Kenya (15%)[52], and contrasts with lower prevalence observed in Europe (France, 12.8%;)[24] and the Middle East (Lebanon, 19.1%). As

maternal samples were obtained within 72 h of NNU admission following delivery at different health facilities, we cannot speculate on the source of maternal carriage which may reflect widespread community prevalence or health facility related acquisition during labour. Dissemination of ESBL-GNB in African communities is widespread, with outpatient cross-sectional studies indicating 32.6% carriage prevalence for children in Guinea-Bissau[51], 21.1% prevalence for children in Madagascar[56] and 63.3% carriage prevalence in adults and children in Egypt[57]. Maternal acquisition of GNB temporally related to hospital admission has also been described with RV carriage prevalence increasing from 18.8% pre-delivery to 41.5% at time of postnatal ward discharge in Sri Lanka[58].

A key finding is that our neonatal cohort carried genetically different MDR-GNB isolates on their skin and intestine compared to paired mother's RV samples. We identified only one newborn-mother pair with identical *E. coli* and *K. pneumoniae* strains, despite high maternal and newborn carriage prevalence. This suggests that mothers do not play a prominent role in newborn MDR-GNB acquisition during the perinatal and early post-natal period at this site. This contrasts with evidence from HIC and MIC indicating that maternal MDR-GNB carriage is a risk factor for neonatal acquisition, with estimated 19% (pooled from 5 studies) of neonatal colonisation associated with transmission from colonised mothers[24]. However, there is a lack of robust genomic research examining mother to newborn MDR-GNB transmission in LIC settings, especially Africa[24] and extrapolation of findings from other settings should be avoided, as neonatal MDR-GNB transmission is complex and influenced by multiple context-specific health system factors[11]. A cross-sectional study of Gambian newborns with clinical early onset sepsis reported low prevalence of vertical transmission from maternal genital tract colonisation with only 14% transmission risk for *S. aureus* and no genotypically related GNB isolates from mother-newborn pairs[25]. A cohort study in South Africa reported similarly low instances (1.1%) of clonal relatedness between maternal and neonatal derived ESBL-producing *Enterobacter cloacae*[59]. A Sri Lankan paired cohort study reported 0.6% maternal transfer rate for ESBL-producing Enterobacterales[58]. A detailed genomic transmission study conducted in a similar low resource NNU in Madagascar also reported no involvement of family members, including mothers, in transmission[60].

We identified no clonal dissemination of *E. coli* and *K. pneumoniae*, suggesting multiple sources. Environmental contamination of NNUs is well recognised[20] with MDR-GNB able to survive prolonged periods on hands[20], medical products such as gastric feeding tubes[61], suction machines[62] water supplies, sinks and inanimate surfaces[18,63]. Endemic and epidemic MDR-GNB outbreaks occurred at our study site in the year preceding this study, with *Burkholderia cepacia* and ESBL-*K. pneumoniae* isolated from IV fluid preparations and antibiotic vials with genotypic linkage to invasive isolates[22]. As environmental samples were not collected during our study we cannot comment on exact sources of environmental acquisition. However, the absence of evidence for mother to newborn transmission, extensive carriage of MDR-*K. pneumoniae* at 7 days and heterogeneous diversity of strains identified is highly suggestive of multiple environmental sources. This should be confirmed by future research, ideally with linked environmental surveillance from the range of sites at which mothers and newborns are managed during labour and the early postnatal period, including place of delivery, referral site and tertiary NNUs.

Limitations of this study include sequencing of single bacterial colonies, which may not have captured the extensive within-host diversity of intestinally carried GNB[64]. Samples were collected over a short period during the dry season and MDR-GNB carriage may differ with seasonality, as shown by other Gambian studies of bacterial infection and carriage[65]. Samples were limited from 14 days onwards due to high mortality. Hence, we are unable to comment on the persistence or resolution of MDR-GNB carriage beyond 7 days nor timing of acquisition of maternal carriage. Our findings are generalisable to similarly low-resource hospital settings and, due to contextual differences in hospital care and environmental bacterial burden, should not be extrapolated to non-comparable settings.

Further research is required to confirm our findings, with larger sample sizes and linkage to clinical outcomes, including elucidation of the exact timing of acquisition, risk factors for MDR-GNB carriage and association with invasive infections. Our findings suggest that multiple environmental sources play an important role in neonatal MDR-GNB transmission, warranting further targeted study to delineate and identify reservoirs at each point along the newborns journey from place of delivery to NNU. Exploration of maternal acquisition of MDR-GNB carriage within both community and hospital settings is also needed to identify interventions to interrupt the circulation of these important neonatal pathogens.

## Conclusion

Gambian hospitalised small vulnerable neonates have high carriage prevalence of MDR- and ESBL-GNB with acquisition between birth and 7d of admission. Despite high maternal MDR-GNB carriage prevalence we identified only limited evidence supporting mother to neonate transmission. Heterogeneous diversity of *E. coli* and *K. pneumoniae* strains and extensive AMR gene presence indicates multiple environmental sources from delivery site to neonatal unit. More comprehensive genomic studies of neonatal and maternal MDR-GNB transmission are required to fully understand acquisition pathways in a variety of low resource settings, to inform development of targeted infection prevention control interventions for the most vulnerable newborns.

## Data availability

Genomic datasets used in this study can be accessed from the Sequence Read Archive, accession number PRJNA73082. For patient confidentiality reasons, access to the linked clinical metadata will be made available following reasonable request to the corresponding author, in line with institutional review board requirements for data sharing.

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

## Acknowledgements

The authors acknowledge the following persons at MRC Unit The Gambia at LSHTM: Yusupha Njie, Binta Saidy and Bai Lamin Dondeh (Data Management); Alpha Jallow, Njilan Johnson, Marie-Rose Thorpe, Elizabeth Batchilly (Research Support). We thank Buntung Ceesay, Mamadou Jallow and Dawda Cham for laboratory contributions and support in addition to Demba Sanneh and Mathurin Diatta (Biobank). In addition, we appreciate the Gambian Government Ministry of Health and Medical Advisory Board at Edward Francis Small Teaching Hospital, for facilitating the data collection. Finally, we would like to thank the newborns and their mothers for generously taking part in this study. The Wellcome Trust (Ref.200116/Z/15/Z) funded collection of samples and microbiological analysis as part of fellowship funding to HB. Grand challenges exploration grant (Ref.OPP1211818) funded genomic analysis. The funders played no role in study design, conduct, analysis or writing of this manuscript.

## Author contributions

H.B. and T.dS. conceptualised the study and obtained funding with input from J.E.L. and B.K.; Data and sample collection were conducted by B.F.K.K. and R.B. with oversight from H.B. and J.E.L.; N.K., M.A.K., and S.D. performed microbiological processing. M.A.K. conducted all DNA extraction and sequencing procedures with input from A.K., Td.S. and A.K.S.; S.Y.B. performed all bioinformatic analyses, including generation of phylogenetic trees and MLST analysis. S.Y.B., M.A.K., and H.B. drafted the manuscript and generated the figures with input from S.D. and Td.S.; All authors provided input to the overall direction and content of the paper and have seen and approved the final version. S.Y.B. and M.A.K. contributed equally to this work.

## Competing interests

The authors declare the following competing interests: B.K. reports grants from the MRC UK Research & Innovation (UKRI), United Kingdom; Wellcome Trust, United Kingdom and Bill and Melinda Gates Foundation (BMGF), United States for a variety of projects related to vaccines and maternal/newborn health. B.K. attended the Gates Global Challenge Meeting in 2022, supported by BMGF, and is also on the Data Safety Monitoring Board for a COVID producing vaccine company. Td.S. is an Editorial Board Member for *Communications Medicine*, but was not involved in the editorial review or peer review, nor in the decision to publish this article. The other authors have no competing interests.
