## [Peer Review File · Communications Medicine]

Reviewers' comments:

Reviewer #1 (Remarks to the Author):

Early acquisition and carriage of genetically diverse multi-drug resistant gram-negative bacilli in hospitalised small vulnerable newborns in the Gambia

Saikou et al

January 2023

Adam Irwin

General comments and recommendation:

Thank you for the opportunity to review this interesting genomic analysis of gram-negative carriage in neonates in Gambia. The manuscript is well written, and though a small dataset, adds new knowledge which support an understanding of the mechanisms of neonatal carriage of gram-negatives in this setting.

Major comments:

Overall:

This small dataset provides genomic data to support a limited role for maternal transmission in the carriage of gram-negatives in neonates. It could be highlighted however that in 1 of only 20 dyads there was likely transmission of the highly virulent and resistant epidemic ST131 E coli.

The study emphasises 'genomic MDR' as 3 or more known resistance genes. Is this conventional? What is the implication of the carriage of 3 beta-lactamases if they confer resistance to the same antimicrobial (or class)? With this in mind, the phenotypic data presented are ambiguous. There is no reference to phenotypic methods for antimicrobial susceptibility testing, and with the exception of Figure 3, little reference in the results. Are these inferred?

There is very limited information to explain the clinical presentations of these infants. The mortality is alarmingly high, but there is only limited information to explain their course. To what extent was infection (and specifically sepsis) implicated? It might be assumed that there was no culture-confirmed infection in the cohort?

The repeated description of study participants as 'small, vulnerable' (including in the title) seems unnecessary. The inclusion criteria only make reference to birthweight, and there is no requirement to be otherwise 'vulnerable' so this seems emotive.

Methods:

The samples were shipped in transport medium, it seems to London. This could be articulated more clearly. Are there any implications for the range of organisms identified? It would also seem appropriate to include the city in which the hospital is based, rather than country alone.

Results:

Line 217 and elsewhere it would be preferable to refer to 'sepsis' rather than 'septic' risk factors

Information regarding the clinical presentations would be valuable if possible.

Figure 1 is extremely helpful. A minor observation that the number of peri-anal swabs totals 57 rather than the stated 56

There is a great deal of valuable information in the description of the identified organisms in neonates and mothers, and their distribution according to sampling site and resistance. It can be difficult to follow however might be usefully summarised in another figure, similar to Figure 1

Discussion:

Line 403 carriage at admission is reported as 44% rather than 41%

Minor comments:

Line 102 I am not sure what the statement "where the bacterial burden differs" means

Line 357 typo in the proportion of AG resistance

Reviewer #2 (Remarks to the Author):

This paper is of great interest as it covers an area that needs to be investigated-how do neonates get colonised with MDR Gram negatives? At what time does this happen after birth? What role does the mothers recto-vaginal flora play in the colonisation of neonates? These are important questions which will provide insights into interventions that would hopefully prevent invasive infections & thus reduce morbidity & mortality in this vulnerable group. The authors found a high carriage rate of MDR which occurred within a week of life. These were not genetically linked to the mother's recto-vaginal flora. There was also great diversity of *E. coli* and *K. pneumoniae* strains which implies multiple environmental sources. Furthermore, multiple AMR genes were present in each isolate.

Although the paper would appeal mostly to paediatricians and neonatologists in particular, it would also generate great interest for infection control practitioners, infectious disease clinicians, physicians, microbiologists etc

The authors conducted a sound investigation, using appropriate investigations which were clearly explained and can be replicated. The results were very interesting & surprising in some aspects. It shed misconceptions on some pre-conceived ideas on how & how soon neonates become colonised with MDROs. The strength of the paper is the sequencing & determination of genetic relatedness among the isolates.

There is research in that has been conducted in this field involving African countries & other resource-limited countries eg Sri Lanka but these have not addressed the same questions. A similar paper that also looked at environmental samples as well, was published recently with

different results. Perhaps the authors can include this in the discussion (Front. Cell. Infect. Microbiol., 25 August 2022 Volume 12 - 2022 | <https://doi.org/10.3389/fcimb.2022.892126>)

The authors have adequately discussed the limitations of their study, which unfortunately are difficult to overcome.

In summary, this is a very interesting paper which provides valuable insights into the acquisition & transmission of MDROs in neonates

Some specific comments:

Line 42 - should read beta lactamases rather than beta lactams

Lines 193-194 - MDR should be defined more clearly especially the differences in ESBL vs MDR. My interpretation is that all ESBLs would be classified as MDR, according to the authors definition but not all MDRs were necessarily ESBL producers. This impacts on lines 281-283...need clarity. If 13 mothers had ESBL producers, how can 16 E.coli be ESBL?

Line 257 - E.coli should be 34/45 rather than 34/46

Line 357 - should be 111 rather than 11

Line 403 - should read 41% rather than 44%, if I understand the figures correctly

Need to correct "gram" to "Gram" throughout

Some sentences are too long & need to be split so they are easier for the reader to follow.

Figure 1 - Total neonatal swabs should be 57 peri-anal & not 56 to give a total of 114

Response to referees

Many thanks to both reviewers for their time and expertise. We have updated the manuscript and figures in response to their comments, as outlined below:

Reviewer 1:

Major comments

1. It could be highlighted that in 1 of only 20 dyads there was likely transmission of the highly virulent and resistant ST131 *E coli*

Many thanks for highlighting the importance of ST131 *E coli* transmission within dyads in our cohort. We recognize that *E coli* 131 strain is an important neonatal pathogen and had already mentioned this along with the finding of low carriage prevalence in the discussion. However, we acknowledge that the specific finding of mother to child ST131 transmission could be emphasized to the reader as it may have relevance for settings with higher *E coli* 131 carriage. We have edited this section of the discussion (page 12, lines 401-403).

2. The study emphasises 'genomic MDR' as 3 or more known resistance genes. Is this conventional? What is the implication of the carriage of 3 beta-lactamases if they confer resistance to the same antimicrobial (or class)? With this in mind, the phenotypic data presented are ambiguous. There is no reference to phenotypic methods for antimicrobial susceptibility testing, and with the exception of Figure 3, little reference in the results. Are these inferred?

Thank you for highlighting the important issue of how genomic MDR is defined. We agree that, as initially written, the definition provided in the methods was not clear and did not adequately describe the classification we used. Genomic MDR was defined as presence of AMR genes encoding for 3 or more different antimicrobial classes. This is a standard definition. Thus, in event of an isolate having 3 beta-lactamase genes, it would be classed as being MDR only if genes conferring resistance to two other antibiotic classes were present. The manuscript has been updated to make this more clear (Page 5, lines 195-197).

Phenotypic resistance data was generated however due to space restriction and main focus of the manuscript being on mother-baby relatedness of isolates and genotypic resistance, we removed it from the methods and results section. We have updated figure 3 to remove reference to phenotypic data.

3. There is very limited information to explain the clinical presentations of these infants. The mortality is alarmingly high, but there is only limited information to explain their course. To what extent was infection (and specifically sepsis) implicated? It might be assumed that there was no culture-confirmed infection in the cohort?

We agree with reviewer 1 that the in-hospital mortality for the cohort was very high. Unfortunately we have limited information about the cause of death and contribution of infections, as blood cultures, CRP, procalcitonin were not available at the study site during this period and it was beyond the remit of the study to investigate cause of death. In addition, post-mortem examinations were not routinely performed as part of standard care. The absence of diagnostic tests was already mentioned in the methods, but this has been emphasised in response to this comment (page 4, lines 142-144). The primary aim of this study was to characterise MDR-GNB carriage in this population, with no objectives related to understanding the clinical course.

4. The repeated description of study participants as 'small, vulnerable' (including in the title) seems unnecessary. The inclusion criteria only make reference to birthweight, and there is no requirement to be otherwise 'vulnerable' so this seems emotive.

Response to referees

We respect the opinion of Reviewer 1 that use of the term 'vulnerable' is not necessary and comes across as emotive. However, low birth weight is a well-established indicator of newborn vulnerability and this umbrella term was used in the manuscript as it encompasses the different newborn phenotypes (preterm, term growth restricted) present in our study population. The Lancet journal is planning to publish "The Small Vulnerable Newborn Series" in 2023, which will report a series of papers to describe different small vulnerable newborn phenotypes, provide updated estimates and characterise adverse outcomes, including neonatal infections, associated with varying phenotypes. With the publication of this series it is expected that the terminology around small newborns will change and we wanted to reflect that by using the term "Small vulnerable" in the title and manuscript.

5. The samples were shipped in transport medium, it seems to London. This could be articulated more clearly. Are there any implications for the range of organisms identified? It would also seem appropriate to include the city in which the hospital is based, rather than country alone.

All samples were processed at MRC Unit The Gambia, with no overseas shipping. This is already described on page 5, lines 176 and 181 and has been articulated more (line 177), as requested. As samples underwent microbiological processing within a short time period with rigorous storage and processing procedures there are no implications for the range of organisms identified. The Gambia is a very small country, with only one teaching hospital, hence why the city was not given, but we have added this into the manuscript (page 4, line 127-128), as suggested by reviewer 1.

6. Line 217 and elsewhere it would be preferable to refer to 'sepsis' rather than 'septic' risk factors
The authors thank reviewer 1 for this comment and have changed all text to "sepsis risk factors" (line 222, Table 1)

7. Figure 1 is extremely helpful. A minor observation that the number of peri-anal swabs totals 57 rather than the stated 56

Many thanks for feedback on this typo. Figure 1 has been updated to correct this.

8. There is a great deal of valuable information in the description of the identified organisms in neonates and mothers, and their distribution according to sampling site and resistance. It can be difficult to follow however might be usefully summarised in another figure, similar to Figure 1

Thank you for this constructive suggestion that another figure may add clarity to the description of organisms by sampling site and resistance. We have added in Supplementary Figure 1, which clearly describes the bacteria detected by sampling site and day of sampling, with 2 additional panels describing distribution of MDR- and ESBL-producing organisms.

9. Line 403 carriage at admission is reported as 44% rather than 41%

Thank you for picking up this typo, which has now been changed to reflect the findings reported in Table 2 (page 13, line 381).

Minor comments:

10. Line 102 I am not sure what the statement "where the bacterial burden differs" means

Thank you for highlighting that this term is not obvious to the reader. We have edited this sentence to state that there are differences in infection prevention control and health system contexts differs in HIC and LMIC settings, rather than bacterial burden (page 3, lines 102-103).

11. Line 357 typo in the proportion of AG resistance

This typo has been amended (page 11, line 329).

Response to referees

Reviewer 2:

1. *A similar paper that also looked at environmental samples as well, was published recently with different results. Perhaps the authors can include this in the discussion*

We are grateful to reviewer 2 for highlighting this recent, relevant study. We have included reference to it in the discussion, particularly with relation to the lower rate of carbapenem resistant organisms (Page 12, line 417-418) and higher prevalence of maternal MDR carriage (page 13, line 426-427) in our study.

2. *Lines 193-194 - MDR should be defined more clearly especially the differences in ESBL vs MDR. My interpretation is that all ESBLs would be classified as MDR, according to the authors definition but not all MDRs were necessarily ESBL producers. This impacts on lines 281-283...need clarity. If 13 mothers had ESBL producers, how can 16 E.coli be ESBL?*

Thank you to reviewer 2 for raising this important issue of how MDR and ESBL are defined in the manuscript. We agree that the initial definition was not clear and we have updated this in the text to state: "Genotypic MDR was defined as presence of AMR genes encoding for three or more different antimicrobial classes, as per MEGARes database" (Page 5, lines 195-197). According to our definitions, it is possible for an isolate to be ESBL-producing but not MDR, if no other AMR genes are present in that isolate. We also agree that the initial manuscript lacked clarity around how we classified AMR genes as being ESBL. We have now explained this in detail (page 5, lines 197-199) and have re-classified isolates as being ESBL-producing in accordance with a published and widely used classification system (Bush-Jacoby, 2010). Table 2 and the corresponding text was also updated with these revised results. We hope this has adequately addressed your comment.

3. *Highlighting of typos, as indicated below:*

Line 42 - should read beta lactamases rather than beta lactams

Line 257 – E. coli should be 34/45 rather than 34/46

Line 357 - should be 111 rather than 11

Line 403-should read 41% rather than 44%, if I understand the figures correctly

Need to correct "gram" to "Gram" throughout

Figure 1 - Total neonatal swabs should be 57 peri-anal & not 56 to give a total of 114

Many thanks for highlighting typos present in the manuscript. These have all been corrected in the revised manuscript

4. *Some sentences are too long & need to be split so they are easier for the reader to follow.*

Many thanks to reviewer 2 for commenting that some sentences are long and could be split to aid readability. In the absence of any specific feedback, we have edited parts of the results and discussion sections to reduce sentence length.

REVIEWERS' COMMENTS:

Reviewer #1 (Remarks to the Author):

The authors have addressed my comments clearly. My only final observation is that the conclusion of the abstract “no evidence for mother’s role in transmission” is more definitive than that of the main conclusions “we identified only limited evidence supporting mother to neonate transmission”. I think this statement more accurately summarises the study.

Reviewer #2 (Remarks to the Author):

I have reviewed the manuscript and rebuttal letter. I am satisfied with the responses & changes that have been made by the authors.

I feel that the article should be published

8th May 2023

Dear reviewers,

Many thanks from the author group for taking the time to re-review the edits and responses to your previous comments. We have addressed the comment from reviewer 1 as detailed below:

“My only final observation is that the conclusion of the abstract “no evidence for mother’s role in transmission” is more definitive than that of the main conclusions “we identified only limited evidence supporting mother to neonate transmission”. I think this statement more accurately summarises the study.”

We agree with this observation by reviewer 1 and have changed the conclusion of the abstract to more accurately reflect the main conclusion.

Yours sincerely

Dr Helen Brotherton on behalf of the author group